# EgoSim: Egocentric Exploration in Virtual Worlds with Multi-modal Conditioning

**Wei Yu**[1,2], **Songheng Yin**[3], **Steve Easterbrook**[1], **Animesh Garg**[2,4,5]
[1]University of Toronto, [2]Vector Institute, [3]Columbia University [4]NVIDIA [5]Georgia Tech

## Abstract

Recent advancements in video diffusion models have established a strong foundation for developing world models with practical applications. The next challenge lies in exploring how an agent can leverage these foundation models to understand, interact with, and plan within observed environments. This requires adding more controllability to the model, transforming it into a versatile game engine capable of dynamic manipulation and control. To address this, we investigated three key conditioning factors: camera, context frame, and text, identifying limitations in current model designs. Specifically, the fusion of camera embeddings with video features leads to camera control being influenced by those features. Additionally, while textual information compensates for necessary spatiotemporal structures, it often intrudes into already observed parts of the scene. To tackle these issues, we designed the Spacetime Epipolar Attention Layer, which ensures that egomotion generated by the model strictly aligns with the camera's movement through rigid constraints. Moreover, we propose the CI2V-adapter, which uses camera information to better determine whether to prioritize textual or visual embeddings, thereby alleviating the issue of textual intrusion into observed areas. Through extensive experiments, we demonstrate that our new model **EgoSim** achieves excellent results on both the RealEstate and newly repurposed Epic-Field datasets. For more results, please refer to `https://egosim.github.io/EgoSim/`.

## 1 Introduction

The success of diffusion models Ho et al. (2020) has revolutionized the field of generative models, enabling advancements from realistic image generation Rombach et al. (2022) to consistent video generation Blattmann et al. (2023b). Recent works on long-term video generative pre-training OpenAI (2024) have further shown that diffusion models can effectively capture the complex dynamics of the physical world, laying a strong foundation for developing world models Ha & Schmidhuber (2018) with practical value.

The next challenge is to investigate how an agent can leverage this foundational model for understanding, interacting with, and planning within observed environments. To this end, we need to first identify what conditioning factors are necessary for constructing an effective world model. Our ultimate aim is to empower an agent to visualize potential scenarios based on observed environmental data, much like playing a game, thereby enhancing its ability to predict and respond to different situations Yang et al. (2023).

**Conditioning with Camera, Frame and Text**: To create a playable simulation engine, the primary desired feature is to enable agents to freely explore the simulated world. This necessitates using the agents' egomotion data, e.g. camera poses, as conditions to guide the generation process of the video diffusion model. Fortunately, as the majority of pixel changes in the video stem from the observer's egomotions rather than dynamic alterations in the surrounding environment, the pre-trained video diffusion models have already successfully internalized prior knowledge about typical patterns and transformations in the 3D world Blattmann et al. (2023b); Voleti et al. (2024). Therefore, our task is to determine how to efficiently extract this prior knowledge and ensure that the videos generated by the model precisely align with the specified camera motion instructions.

Existing methods Wang et al. (2023); He et al. (2024) typically combine the transformed camera information directly with intermediate features before feeding them into the temporal transformer

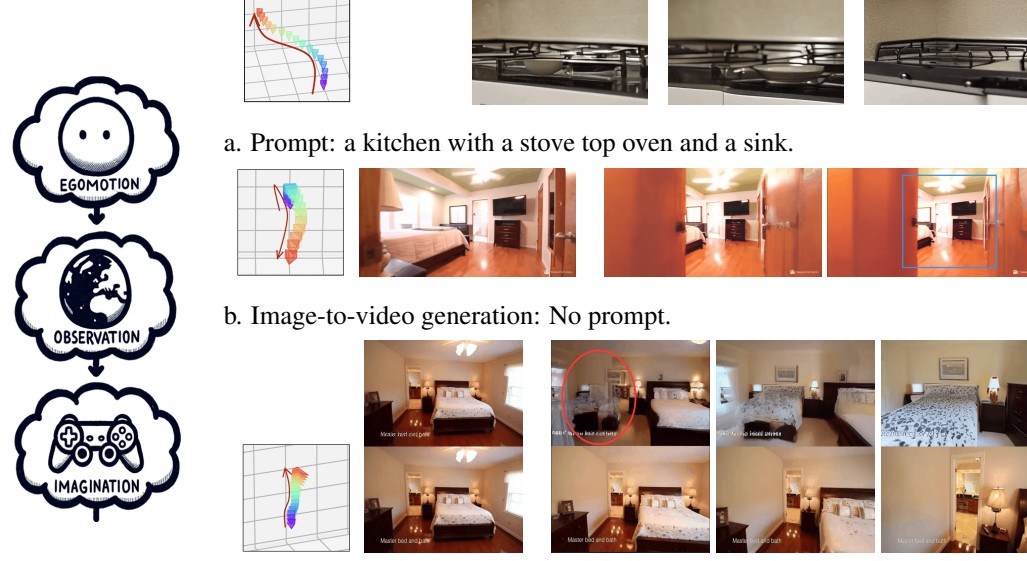

a. Prompt: a kitchen with a stove top oven and a sink.

b. Image-to-video generation: No prompt.

c. Prompt: a bedroom with a bed and dresser in it.

Figure 2: To create a playable world simulator that supports free exploration and imagination based on observation, we have identified several primary bottlenecks: (a). We attempt to generate a video of moving forward in front of a stove, but the model fails to fulfill it because it has never seen such training data of moving above a stove. (b). Image-to-video model equipped with camera control, as emphasized inside the blue box, can only depict details already observed in the first frame. (c). Enable text-to-video models with image conditioning. The first row is a generated video while the second row is the groundtruth. As illustrated inside the red ellipse, introducing textual information can erode the already observed part, leading to a disruptive transition from realistic scenes to complete fantasy.

layer. This approach can indeed roughly learn the camera motion. However, it is quite intuitive to notice that the intermediate features, along with the camera poses, jointly determine how the video is generated. As shown in Figure 2a, influenced by the distribution of its training data, the model may determine that even if the input motion is valid in certain situations, the camera cannot move forward because it has never seen an example of forward movement in such scenarios. Therefore, the generated video in this case will severely deviate from the user-specified motion.

Additionally, we aim for the model to seamlessly integrate observed environmental information, usually presented as context frames, into the video generation process while simultaneously inferring unobserved structures and predicting future interactions with the environment. Our preliminary experiments indicate that although the pretrained image-to-video model Blattmann et al. (2023a) can produce authentic videos with minor movements, increasing the amplitude of the movement results in outputs that only depict the details visible in the context frame, as illustrated in Figure 2b. This suggests that the model needs supplementary information to imagine the unobserved parts.

Moreover, through further analysis on text-to-video models Guo et al. (2023c), we found that adding textual information can effectively compensate for previous shortcomings by injecting extra spatiotemporal structure, leading to more coherent and realistic video sequences. However, as shown in Figure 2c, it became evident that this added spatiotemporal structure intrudes upon already observed parts, causing a disruptive transition from realistic scenes to complete fantasy.

In summary, our detailed examination reveals that while each condition in multi-condition inputs has an effective controlling method, *integrating these conditions together often leads to their embeddings interfering with or negating each other, significantly undermining the reliability of the generated video*. This indicates that using a naively unified embedding control interface may inevitably harm the accuracy of video generation. Consequently, this paper focuses on **resolving the interactions between different conditioning modalities to construct a coordinated and compositional world model**. The contributions of our paper can be categorized as follows:

1. We introduce a novel plug-in DiT Peebles & Xie (2023) module called the **S**pacetime **E**pipolar **A**ttention layer (SEAL) to override the interference of other features in video generation, making it more accurately aligned with camera motion.

2. We propose CI2V-adapter to arrange text information and visual conditions in a mutually exclusive relationship. CI2V-adapter leverages camera movement information to further assist in clearly defining the boundaries between the text and visual elements, ensuring they do not interfere with each other during the generation process.

3. We repurpose the Epic-Field dataset for egocentric video generation and established a new benchmark to evaluate video diffusion models in more interactive and dynamic settings.

4. We evaluate the proposed method **EgoSim** on two competitive benchmarks, RealEstate and Epic-Field datasets, in a multi-condition-input setting. Extensive experimental results show that our model achieves precise control that previous methods could not accomplish. It not only generates video that accurately follows camera movements but also fills in unobserved new information into existing observations and interacts with the environment.

## 2 ORCHESTRATING DIVERSE CONDITIONS

Given a reference image, a text prompt and a sequence of camera poses, our goal is to generate a video sequence which starts from the context frame, faithfully obeys the user-specified motion and interact with environment in accordance with the textual description. As discussed earlier, the current control methods for these conditioning factors often interfere with each other, leading to the generation of videos that are inconsistent and unexpected.

Our initial analysis has pinpointed two main sources of this conflict: the interaction between camera control and intermediate features, and the interplay between text and context frames. This chapter will begin by providing an overview of video generation models and then address these two critical issues separately to resolve the problems.

### 2.1 PRELIMINARY

Diffusion Models Ho et al. (2020) are a class of generative models designed to produce high-quality samples through a multi-step process. Starting with Gaussian noise, these models iteratively refine and denoise the initial random input. In the case of video generation Blattmann et al. (2023b), a sequence of $N$ images (or their latent features) $z_0^{1:N}$ are progressively subjected to noise $\epsilon$, transforming them into a normal distribution over $T$ steps. A neural network $\epsilon^{\sigma}$ is then trained to predict the added noise from these noised inputs. During training, the network aims to minimize the mean squared error (MSE) between its predictions and the actual noise. The training objective function is defined as follows:

$$\mathcal{L}(\theta) = \mathbb{E}_{z_{0:N}, \epsilon, c_t, t} \left[ \left\| \epsilon - \hat{\epsilon}_\theta(z_t^{1:N}, c_t, t) \right\|_2^2 \right]$$

where $c_t$ represents conditional embeddings.

### 2.1.1 CONTROLLABLE VIDEO GENERATION

Controllability plays an important role in video generation as it enables users to craft content precisely as they envision. In this work, we aim to integrate three distinct control conditions— **autonomous camera motion, reference images, and text** —into a single pipeline.

First, regarding camera motion, it's important to note that the field of video diffusion models is still relatively new. Control over camera motion has only been tentatively explored so far. Recent methods Wang et al. (2023); He et al. (2024) primarily achieve control by combining camera embeddings with intermediate features. On the other hand, there is a wealth of research on both text-to-video Guo et al. (2023c) and image-to-video generation Blattmann et al. (2023a), but combining these two approaches remains rare. Typically, text-to-video generation incorporates CLIP embeddings Radford et al. (2021) of texts through multiple cross-attention layers while image-to-video models are mainly trained from scratch by concatenating repeated first-frame image features to noised input.

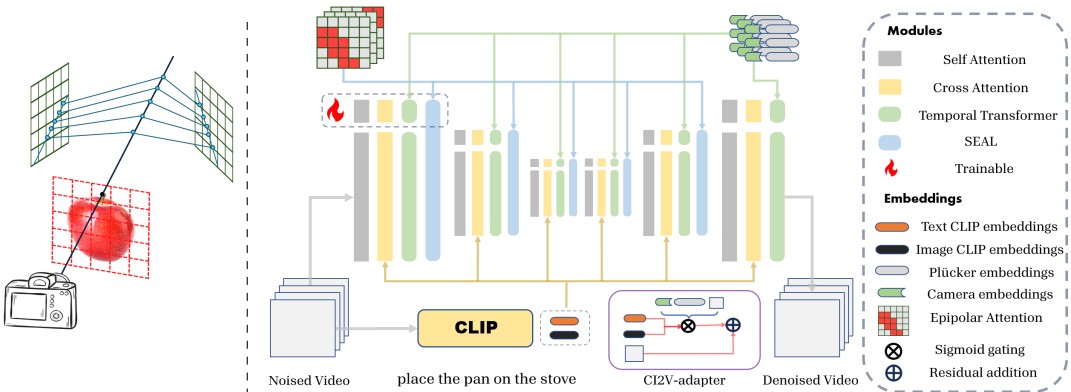

Figure 3: **EgoSim Overview**. Left: A simple illustration of how epipolar attention is calculated. Each patch only attends to patches from other frames which directly intersect or are near the re-projection of the unprojected epipolar line. Right: This diagram shows how various embeddings are injected into the video diffusion model. All modules along with the entire SEAL inside the dashed box with a flammable sign are set trainable. The computation graph of the CI2V-adapter is shown within the purple box. For simplicity, ResNets are omitted and trainable parts are not repeatedly emphasized with flammable sign for other blocks.

## 2.2 CAMERA CONTROL

In this paper, we attempt to develop a more robust and reliable method for controlling camera trajectories in video generation. Existing approaches, which concatenate or add camera embeddings with intermediate features Wang et al. (2023); He et al. (2024), often result in camera movements being influenced by these features. Therefore, we need a control method that operates relatively independently of these features and enforces rigid constraints derived from camera movement.

### 2.2.1 SPACETIME EPIPOLAR ATTENTION LAYER

We notice that UNet-based diffusion models have an inherent limitation in egocentric video generation. Specifically, current publicly available pre-trained models all utilize factorized space-time attention, meaning a patch in one frame cannot directly attend to a patch in another frame. This design helps save computational resources but leaves frame-to-frame consistency solely controlled by the text prompt. Introducing additional camera information, which governs egomotion, is likely to disrupt the control provided by the textual guidance. Therefore, it becomes necessary to introduce a mechanism that enables direct interaction between patches across different frames.

Inspired by SORA OpenAI (2024), we addressed this limitation by first rearranging the latent representation into spacetime patches and introducing the DiT Peebles & Xie (2023) structure to enable interaction between patches. Moreover, we propose to leverage geometric constraints derived from epipolar lines to precisely guide camera movements. The use of epipolar attention introduces additional sparsity, enabling us to utilize memory-efficient operations. We call this module Spacetime Epipolar Attention Layer (SEAL).

Next, we will describe in detail how to utilize the newly established channel to enforce camera control by leveraging the epipolar geometry. Consider a patch coordinate $(u, v)$ in the target frame $I_t$, where the intrinsic parameters $\mathcal{K}_t$ and extrinsic parameters, including rotation $\mathcal{R}_t$ and translation $\mathcal{T}_t$ relative to canonical frame, are known. The epipolar line $\mathbf{l}_i$ corresponding to this patch can be calculated using the fundamental matrix:

$$\mathbf{l}_i = \mathcal{F}_i \left[u, v, 1\right]^T = \mathcal{K}_i^{-T} \left([\mathcal{T}_t]_\times \mathcal{R}_t\right) \mathcal{K}_t^{-1} \left[u, v, 1\right]^T \tag{1}$$

As illustrated in Figure 3, with the help of epipolar lines, model can identify which regions in other frames to focus on for any given patch as follows:

$$\text{EpipolarAttention}(Q, K, V) = \text{softmax}\left(\frac{QK_\star^T}{\sqrt{d_k}}\right) V_\star \tag{2}$$

where $K_\star$ and $V_\star$ are keys and values calculated from patches directly crossed by or near the projection of epipolar lines. After constructing such links between all patches, the model can learn to effectively extract knowledge about viewpoint changes in the 3D world from video generative pre-training.

### 2.2.2 INJECTION OF CAMERA EMBEDDINGS

Since the temporal transformer mainly focuses on learning motion information, integrating the camera embedding Wang et al. (2023); He et al. (2024) into the temporal transformer can achieve the most direct impact on egomotion generation. More importantly, it is compatible with epipolar attention. Therefore, this paper retains and improves the method of injecting camera embedding.

Methods like MotionCtrl Wang et al. (2023), which directly flatten the camera pose, can also capture camera motion but overlook intrinsic parameters. Similar to a concurrent work, CameraCtrl He et al. (2024), we concatenate the camera positional encoding Vaswani et al. (2017) with Plücker embedding Sitzmann et al. (2021) to get a more precise geometric interpretation for each patch. However, unlike CameraCtrl, we found that introducing an additional encoder to generate multi-scale camera embeddings is unnecessary. Instead, we employ pixel unshuffle Shi et al. (2016) to adjust the size while preserving as much fine-grained positional information as possible. This approach is sufficient and also helps to save computational resources.

### 2.3 FRAME V.S. TEXT

As previously mentioned, after we equipped the video model with camera movement control, relying solely on the context frame was insufficient for the model to envision the details of unobserved scenes. This underscores the necessity of text input, which supplements the imagined spatiotemporal structure. However, we also realized that textual information could be intrusive and should therefore be mutually exclusive with visual input. In other words, a particular patch in a specific frame should be explained either by text or by the context frame, but not both. To address this, we notice that the shift between imagination and observation is primarily driven by egomotion. Therefore, incorporating camera motion data is essential to better determine which source of information to prioritize.

### 2.3.1 CI2V-ADAPTER

To preserve the intricate text-to-video capabilities that the cross-attention layers have learned, we decide to start from text-to-video generation model and then adapt it to support image-to-video generation. Specifically, our improvement is built based on the I2V-adapter Guo et al. (2023a). The I2V-adapter pipeline consists of two key components: (a). Additional trainable cross-attention operations performed in each self-attention layer for the first frame where ground truth is provided. (b). IP-adapters Ye et al. (2023) attached after cross-attention layers through adding the output of extra cross-attention computations with image embeddings. We upgrade I2V-adapter into CI2V-adapter by enabling both components to incorporate the influence of camera motion.

For the first component, we notice that it actually involves cross-frame interaction that closely resembles SEAL computation. Hence, we can directly incorporate epipolar attention masks precomputed in SEAL, relative to the first frame, into the cross-attention mechanism. This allows for more precise feature fusion across frames.

For the IP-adapter component, we want to mitigate mutual interference between textual and visual control through providing additional guidance from camera informatiom. More specifically, intermediate features attached with corresponding camera embeddings are used as input for patch-wise sigmoid gating functions $\mu$ to form a weighted sum of two types of information. This integration is calculated as follow:

$$\mu_t = \phi(W_2 * \text{ReLU}(W_1 * [x_t, \mathcal{C}_t] + b_1) + b_2) \tag{3}$$

$$x_t = (1 - \mu_t) \odot \text{X-attn}(\mathcal{T}, x_t) + \mu_t \odot \text{X-attn}(\mathcal{I}_1, x_t) + x_t \tag{4}$$

where $x_t$ is the intermediate feature of frame $t$, $\mathcal{T}, \mathcal{I}, \mathcal{C}$ are transformed text, image and camera embeddings, $\phi$ is sigmoid activation, $*$ is the convolution operator, $\odot$ is the Hadamard product and X-attn is cross-attention operation.

## 2.4 TRAINING DATA

Data is the most critical component of video generation models. Previous work on video generation with camera control has often been limited to datasets featuring static scenes, e.g. RealEstate Zhou et al. (2018), which restricts the model's ability to generalize to more dynamic environments. This is due to the fact that labeling camera poses for dynamic scene data is very challenging and complex. Fortunately, we have identified a highly suitable dataset, Epic-Field Damen et al. (2018), which, although not previously used in video generation research, aligns perfectly with our requirements.

However, the Epic-Field dataset is not ready to be used directly. To transform it into a resource suitable for egocentric video generation, we installed two major and laborious upgrades: (a). we re-annotated the entire dataset's text labels using CogVLM2 Hong et al. (2024), and (b). due to the significant camera motion, we applied NAFNET Chu et al. (2022) to perform motion deblurring on each frame. We applied the same modifications to the RealEstate dataset as well.

## 3 EVALUATION

We will now evaluate the proposed method, EgoSim, and demonstrate its applications across various scenarios. In particular, we will highlight the following key capabilities of the model: (a). precise control of camera movement, and (b). generating meaningful interactions with the environment based on both observation and imagination.

To demonstrate these two capabilities, the experiments are organized into three distinct scenarios: (1) camera + text-to-video (Section 3.2.1), (2) camera + image-to-video (Section 3.2.2), and (3) camera + image + text-to-video (Sections 3.2.3 and 3.3).

## 3.1 EXPERIMENTAL SETUP

**Pretrained weights**: We leverage two pretrained video diffusion model as base to implement our method. More specifically, for the image-to-video case, we use Stable Video Diffusion (SVD) Blattmann et al. (2023a). For experiments that require text input, we use AnimateDiff Guo et al. (2023c).

**Baselines**: To the best of our knowledge, there is no video model that can simultaneously accept camera information, video frames, and text as conditional inputs. Consequently, we can only attempt some simplified experimental scenarios, or we can modify existing methods for comparative purposes. CameraCtrl He et al. (2024) and MotionCtrl Wang et al. (2023) are included as baselines for video diffusion model equipped with camera motion control. Methods which can modify AnimateDiff for image conditioning involve I2V-adapter Guo et al. (2023a) and SparseCtrl Guo et al. (2023b). We reproduced the above models following the respective research papers and publicly available code.

**Metrics**: To estimate the fidelity of egocentric video prediction, SSIM (Wang et al. (2004)) and the Fréchet Video Distance (FVD) Unterthiner et al. (2019) are calculated between the predictions and groundtruths. Besides, we recruit Colmap Sayab et al. to assess the camera poses of generated videos to see if they authentically follow camera movements. **TransErr** and **RotErr** He et al. (2024) are computed by comparing the ground truth camera poses $[R—T]$ and estimation $[R_*—T_*]$, as follows.

$$\text{RotErr} = \sum_{j=1}^{n} \arccos\left(\frac{\text{tr}(R_*^j R^{jT}) - 1}{2}\right) \quad \text{and} \quad \text{TransErr} = \sum_{j=1}^{n} \|T^j - T_*^j\|_2$$

We use a mixture of groundtruth trajectories and more difficult random trajectories to calculate the above errors.

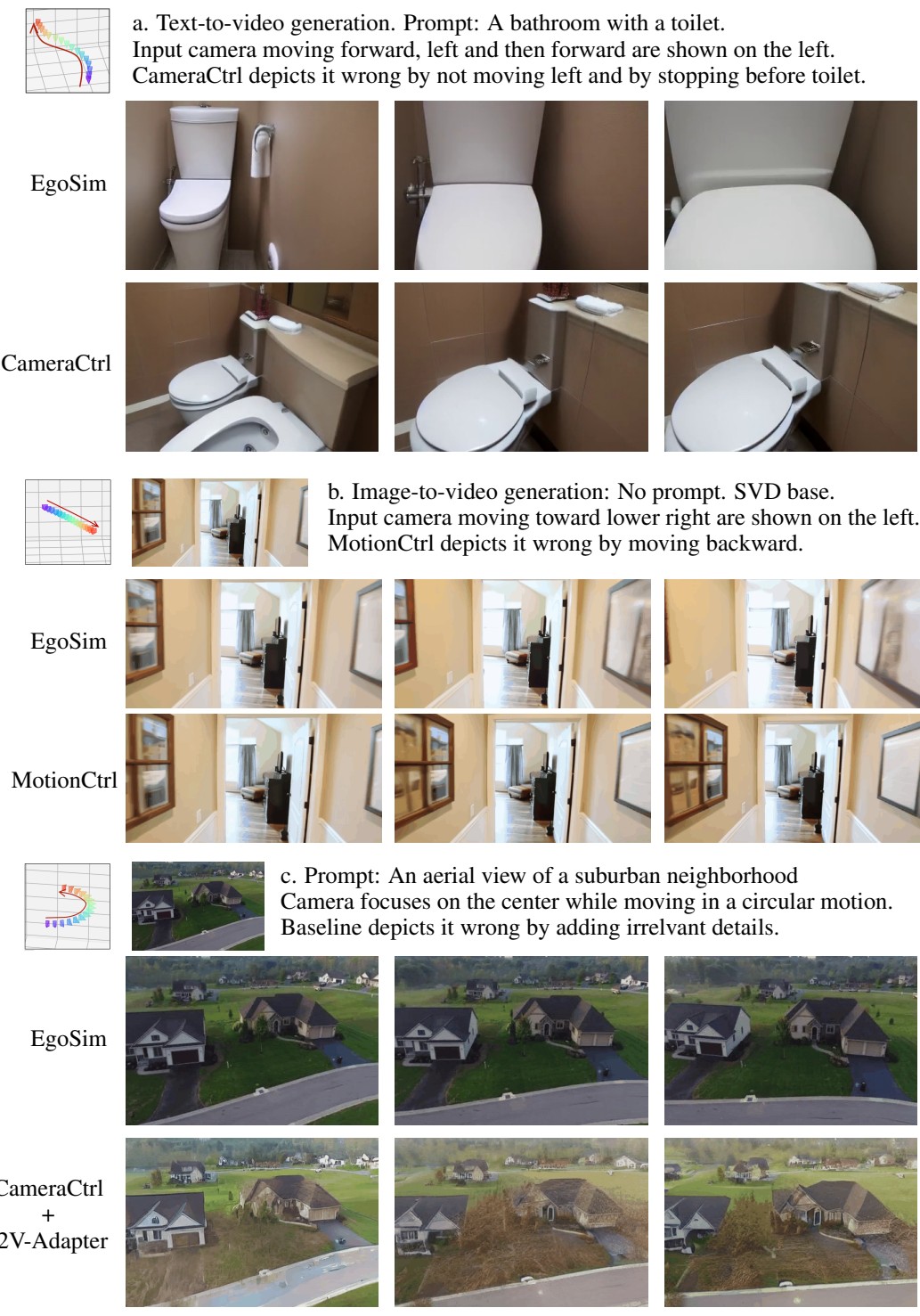

Figure 4: Qualitative Comparison on RealEstate Datasets.

## 3.2 REALESTATE

RealEstate Zhou et al. (2018) consists of a large number of open house video tours. In the case of AnimateDiff, we resize the video resolution to $256 \times 384$. We train and test the model with a length of 16 frames. For SVD, we resize to $256 \times 512$ and use $T = 14$.

### 3.2.1 TEXT-TO-VIDEO GENERATION WITH CAMERA CONTROL

In this section, we will first evaluate the effectiveness of SEAL for text-to-video model equipped with camera control on the RealEstate dataset. As presented in Table 1, the proposed method outperforms all previous methods by a wide margin on all metrics.

Qualitative analysis in Fig 4a further demonstrates the limitations of previous methods and the superiority of our approach. We can see that the baseline model is unable to generate out-of-distribution movements because its control is influenced by intermediate features. In contrast, our method perfectly adheres to the specified camera movements. In addition, SEAL also improves the consistency and realism of the videos thanks to its spacetime structure.

| Method | TransErr↓ | RotErr↓ | FVD↓ |
|---|---|---|---|
| MotionCtrl | 9.88 | 1.23 | 793.5 |
| CameraCtrl | 9.67 | 1.23 | 782.4 |
| EgoSim | **8.01** | **0.80** | **722.0** |

Table 1: RealEstate-T2V results.

| Method | TransErr↓ | RotErr↓ | FVD↓ | SSIM↑ |
|---|---|---|---|---|
| MotionCtrl | 7.74 | 0.88 | 330.1 | 0.678 |
| CameraCtrl | 6.98 | 0.81 | 318.3 | 0.691 |
| EgoSim (SVD) | **5.21** | **0.53** | **223.8** | **0.831** |

Table 2: RealEstate-I2V results.

| Method | RealEstate-T+I2V | | | |
|---|---|---|---|---|
| | TransErr↓ | RotErr↓ | FVD↓ | SSIM↑ |
| MotionCtrl + I2V | 14.10 | 1.71 | 488.3 | 0.637 |
| CameraCtrl +SparseCtrl | - | - | 1360.8 | 0.353 |
| CameraCtrl + I2V | 13.95 | 1.68 | 472.9 | 0.645 |
| EgoSim | **6.75** | **0.77** | **293.7** | **0.808** |

Table 3: RealEstate-T+I2V results.

| Method | Epic-Field-T+I2V | | | |
|---|---|---|---|---|
| | TransErr↓ | RotErr↓ | FVD↓ | SSIM↑ |
| MotionCtrl + I2V | 16.31 | 1.70 | 1223.5 | 0.612 |
| CameraCtrl +SparseCtrl | - | - | 1566.2 | 0.331 |
| CameraCtrl + I2V | 15.95 | 1.77 | 1172.1 | 0.625 |
| EgoSim | **12.41** | **1.27** | **663.2** | **0.755** |

Table 4: Epic-Field-T+I2V results.

### 3.2.2 IMAGE-TO-VIDEO GENERATION WITH CAMERA CONTROL

Next, we move to the pure image-to-video setting and compare our method with MotionCtrl + SVD and CameraCtrl + SVD. it should be noted that estimating SSIM becomes practically meaningful in the case of frame-conditioned generation because SSIM can more effectively tell us if the model strictly follows the given camera motion for a given groundtruth-generation pair. Besides, as SVD uses more parameters, calculations and data compared to AnimateDiff, the overall generated video quality is better.

**Results**: As shown in Table 2, EgoSim significantly improves on all metrics compared to the baseline methods. In Figure 4b, we can see that because the context frame we provide is a long corridor, MotionCtrl would assume that movement to the lower right corner should not occur even if it's a valid movement. On the contrary, our model can faithfully generate the corresponding movement to the lower right corner as expected.

### 3.2.3 CAMERA, FRAME AND TEXT

As we mentioned earlier, due to the lack of existing baselines in this setting, we can only create baseline models for comparison by combining existing methods. More specifically, we tried the following three combinations: MotionCtrl + I2V-Adapter, CameraCtrl + I2V-Adapter and CameraCtrl + SparseCtrl Guo et al. (2023b). It is worth noting that CameraCtrl + I2V-Adapter in general performed well, but we were unable to successfully reproduce the results with CameraCtrl + SparseCtrl and cannot calculate TransErr and RotErr even though we directly used the publicly available code and the pre-trained weights provided by the authors.

**Results**: Quantitative results are summarized in Table 3 and EgoSim achieves the best performance. In Figure 4c, we can see that CameraCtrl + I2V-Adapter suffers from the intrusion issue while the proposed method can add reasonable imagination on top of observed details.

## 3.3 Epic-Field

Finally, we move to the most challenging setting, Epic-Field Damen et al. (2018). Compared with RealEstate, Epic-Field involves not only camera movements but also a significant amount of interaction with the environment. Thus, the model needs to learn additionally how to generate videos of these actions. We trained an additional LORA for Epic-Field, and as in the previous section, we conducted experiments using MotionCtrl + I2V-Adapter, CameraCtrl + I2V-Adapter and CameraCtrl + SparseCtrl as baselines. We resized the video frames to $256 \times 448$ and use $T = 14$.

**Results**: The quantitative comparisons are provided in Table 4 and EgoSim achieves the best scores on all metrics. The qualitative analysis in Fig 5 further reveals the advantage of our method. We encourage readers to view more impressive visual results in the project page. EgoSim not only learned to generate precise autonomous movements but also can perform environmental interactions such as washing dishes and opening drawers. This significantly broadens the range of potential applications for EgoSim. Past inverse dynamics approaches Du et al. (2024) were limited in their ability to control autonomous movements, restricting test scenarios to fixed camera positions. However, with the advent of EgoSim, we can now attempt to learn more advanced inverse dynamics models. This enables dynamic camera positioning and broader testing capabilities, paving the way for more robust and versatile autonomous systems.

## 3.4 Ablation study

In this section, we evaluate the contribution of each module to the model's improvement through extensive ablation experiments. As shown in Table 5, each modification contributes to either the improvement of the video quality or better accuracy of camera control.

| Method | RealEstate-I2V | | | | Epic-Field | | | |
|---|---|---|---|---|---|---|---|---|
| | TransErr↓ | RotErr↓ | FVD↓ | SSIM↑ | TransErr↓ | RotErr↓ | FVD↓ | SSIM↑ |
| AnimateDiff+I2V+MotionCtrl | 14.10 | 1.71 | 488.3 | 0.637 | 16.31 | 1.70 | 1223.5 | 0.612 |
| + Plucker embedding | 13.96 | 1.68 | 466.5 | 0.644 | 16.08 | 1.77 | 1213.0 | 0.611 |
| + SEAL | 8.12 | 1.01 | 366.7 | 0.739 | 13.81 | 1.41 | 786.2 | 0.698 |
| + CI2V-adapter | **6.75** | **0.77** | **293.7** | **0.808** | **12.41** | **1.27** | **663.2** | **0.755** |

Table 5: Ablation study. Note that for TransErr, RotErr and FVD, lower number indicates better performance while higher SSIM means better.

## 4 Related Work

### 4.1 Video generation

Since the advent of the deep learning era, video generation has been an active area of research. Initially, ConvLSTM model Shi et al. (2015); Wang et al. (2017); Yu et al. (2020) was popular, followed by GAN models Skorokhodov et al. (2022); Yu et al. (2022), and more recently, video diffusion models Blattmann et al. (2023b) have become the mainstream. Video diffusion model Blattmann et al. (2023a) usually extends a 2D image diffusion architecture to handle video data, enabling joint training on both images and videos from the ground up. To leverage powerful pre-trained image generators like Stable Diffusion Rombach et al. (2022), subsequent approaches expand the 2D architecture by integrating temporal layers between the pre-trained 2D layers. This new model is then fine-tuned on a large video dataset. The latest advancement is SORA OpenAI (2024), which abandons the aforementioned strategy and directly uses diffusion transformers Peebles & Xie (2023) to train on large-scale videos of different sizes. Its generation quality is exceptionally impressive.

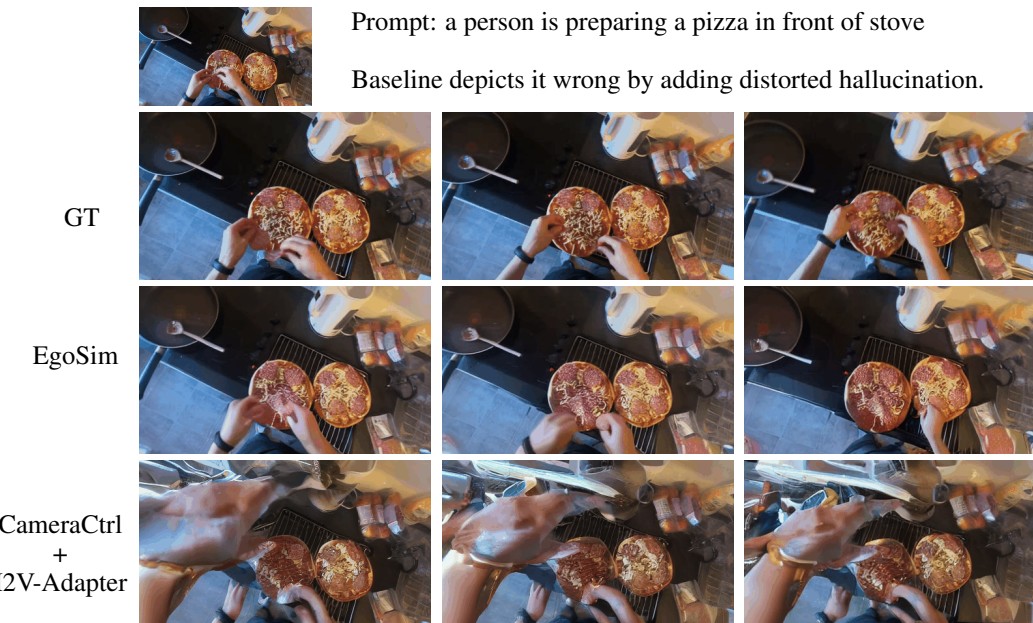

Prompt: a person is preparing a pizza in front of stove

Baseline depicts it wrong by adding distorted hallucination.

Figure 5: Qualitative Comparison on Epic-Field Dataset.

## 4.2 VIDEO GENERATION WITH CAMERA CONTROL

Research on video generation with camera control Yu et al. (2024) is still in its early stages. The first two works, MotionCtrl Wang et al. (2023) and CameraCtrl He et al. (2024), relied on directly concatenating or adding camera embeddings to control camera movement. Researchers soon recognized the limitations of such methods. Epipolar attention Kant et al. (2024); Du et al. (2023) has been commonly studied in the context of 3D generation, but its potential for video generation remains largely unexplored. It is worth noting that two other concurrent works, CamCo Xu et al. (2024) and CVD Kuang et al. (2024), have also acknowledged the importance of incorporating epipolar attention. However, unlike their approaches, our work not only leverages epipolar attention but also, independently of their efforts, identifies and addresses the limitations present in their methods. CamCo focused only on image-to-video (I2V), which our early analysis showed I2V fails to depict significant camera movements, covering only part of our research. CVD, meanwhile, requires generating multiple videos and cannot directly produce a single one. While they recognized the importance of cross-frame interaction, they took an indirect approach by establishing interaction between two different video trajectories.

## 5 CONCLUSION AND FUTURE WORK

This paper introduces EgoSim, a compositional world simulator that can egocentrically explore and interact with the observed environment. To achieve this, we identify two major obstacles that prevent the creation of meaningful videos that accurately adhere to user-specified instructions and tackle them with spacetime epipolar attention and CI2V-adapter. Our designed model has demonstrated unprecedented controllability and the potential uses of such an egocentric world simulator are also diverse and impactful. Here, we summarize two interesting directions for future work:

**DiT video diffusion**: Although our method is plug-and-play, a key challenge arises from the fact that more powerful DiT diffusion models often employ 3D VAE to compress the temporal dimension. This compression complicates the process of the 3D projection used in epipolar attention to directly correspond to specific spacetime patches, making it more difficult to maintain precise attention.

**Inverse dynamics**: The most suitable application scenario for this method is inverse dynamics. We hope to test EgoSim soon for training robots, giving embodied agents the ability to plan and simulate the future through imagination.

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

# A APPENDIX

## A.1 TECHNICAL DETAILS

### A.1.1 DATA

For RealEstate dataset, we use 67,477 scenes for training and 7,289 scenes for testing. For Epic-Field, we use 611 scenes for training and 88 scenes for testing. It should be noted that the duration of each scene of Epic-Field is much longer than that in RealEstate. That's why the number of scenes in Epic-Field is much smaller.

We found that a large number of videos in Epic-Field suffer from significant motion blur due to rapid movement, which is detrimental to model learning. Therefore, we applied NAFNet Chu et al. (2022) to perform motion deblurring on all frames.

During the training, we use sample stride of 6 for RealEstate and 4 for Epic-Field. We also used several additional techniques for data augmentation. In RealEstate, since most scenes are static, we can reverse the videos to generate additional motion trajectories. In both datasets, we can randomly increase or decrease the sample stride by one step to obtain video clips with different speeds. As a result, each training sample is consist of a 14-frame video clip, a text prompt and camera poses for all frames.

### A.1.2 TRAINING AND INFERENCE

In the case of SVD and AnimateDiff, we keep the original parameters of our base models fixed and only optimize the newly introduced layers and their subsequent layers using the AdamW optimizer with learning rate $2 \times 10^{-4}$. All models are trained on 8 NVIDIA A100 GPUs for 300k iterations using an effective batch size 32. We use BF16 precision for training SVD.

We use DDIM scheduler with 1000 steps during training and 25 steps during inference.

### A.1.3 EVALUATION

For calculation of TransErr and RotErr, we basically follow the protocol from CameraCtrl. The main difference is that we adopted different camera poses. We sorted the translation and rotation of the real camera poses from two datasets, and then randomly selected poses from the top 20% to 40% and 40% to 60% ranges to classify them into hard and medium groups, respectively. Additionally, we designed simple linear motions and rotations to create the simple group. We combined the hard, medium, and simple groups in a 1:1:1 ratio to adequately test the camera poses. Therefore, our selection process, which filters out poses with larger motion ranges, results in more challenging evaluation. For FVD and SSIM, we follow the common practice.

## A.2 RELATED WORK

The exploration of egocentric 3D generation through direct camera control has undergone several paradigm shifts. InfiniteNature Liu et al. (2021) stands out as an early influential work in this field. This project primarily utilizes traditional computer vision techniques. It takes a paired RGB image and a disparity map to construct a textured mesh and then renders from new perspectives by warping the textures to adjust disparities. To complete the process, a refinement network addresses and corrects any gaps or missing parts in the final output. This method does not fully leverage the learning capabilities of neural networks.

After the rise of generating images using transformer-based autoregressive models, GeoGPT Rombach et al. (2021) posits that all camera motion control conditions can be transformed into tokens or directly summed embeddings. They experimented with a wide variety of ways for incorporating control conditions to assess their impact on the generated results. However, their fusion methods only included concatenation and addition. Therefore, GeoGPT encounters the same issues as CameraCtrl and MotionCtrl, specifically limited ranges of generated motion. Therefore, subsequent studies primarily focused on how to introduce more explicit geometrical control. "Look Outside the Room" Ren & Wang (2022) proposed using Camera-Aware Bias by computing a bias with a Multi-Layer Perceptron (MLP), which takes the relative camera position as input during the calculation of the

attention score. The paper Du et al. (2023) utilized epipolar attention to aggregate information from patches under different viewpoints to generate the final render. These efforts have achieved significant progress.

Subsequently, the era of diffusion models arrived. Researchers Tseng et al. (2023) introduced epipolar attention into the use of diffusion models for 3D generation. SceneScape Fridman et al. (2024) returned to the framework of InfinitNature, but with the aid of a more powerful diffusion model. It integrated text control capabilities and achieved better results.

