# OpenReview forum: "EgoSim: Egocentric Exploration in Virtual Worlds with Multi-modal Conditioning"
_ICLR.cc/2025/Conference — ICLR 2025 Poster_

### Official Review · Reviewer_L7sN · 2024-11-03

**Soundness:** 2
**Presentation:** 2
**Contribution:** 2
**Rating:** 6
**Confidence:** 4

**Summary:**

This work focuses on improving the controllability of video diffusion models for egocentric exploration with camera ego-motion + text + image as conditioning. The proposed framework incorporates epipolar constraints to better focus on the relevant parts in the attention mechanism, referred to as Spacetime Epipolar Attention (SEAL). It is then combined with existing text-to-video and image-to-video diffusion models. Experiments on RealState and EPIC-Fields datasets show the effectiveness of the proposed approach over existing methods.

**Strengths:**

- This work focuses on improving the controllability of video diffusion models with multiple conditioning factors: combinations of camera ego-motion, text, and image, which is important from a practical use perspective.
- The idea of incorporating epipolar constraints into the attention mechanism is simple and intuitive.
- Experiments on RealState (Tab.1, Fig.4) and EPIC-Fields (Tab.1, Fig.5) datasets show the effectiveness of the proposed approach over existing methods.
- Ablations in Tab.2 and visualizations on the website provide a better understanding of the capabilities of the proposed approach.

**Weaknesses:**

- In Tab.1, several values are missing, which makes it difficult to compare different models. There is some justification in the text (L426-429). It'd be helpful to have more details:
    - Why can't MotionCtrl and EgoSim (SVD) be evaluated on EPIC-Fields? Both are I2V methods and EPIC-Fields contains both frames and camera egomotion.
    - For CameraCtrl + SparseCtrl, why can't TransErr and RotErr be computed? Since ground truth is available, the camera trajectory from the video diffusion output needs to be computed. Is it because COLMAP optimization does not converge on the outputs? Is there some other reason?
    - It'd also be useful to have T2V and I2V settings on EPIC-Fields to better understand the trends across different datasets. Since text description is available for EPIC-Fields (Fig.5), is there any reason to not use these settings?
- There are 2 mentions of efficiency benefits in the text. It'd be helpful to verify these benefits quantitatively, in terms of memory usage and train/inference time.
    - L205-206: The use of epipolar attention introduces additional sparsity, enabling us to utilize memory-efficient operations.
    - L236-238: employ pixel unshuffle Shi et al. (2016) to adjust the size while preserving as much fine-grained positional information as possible. This approach is sufficient and also helps to save computational resources.
- L245-246 mentions: 'a particular patch in a specific frame should be explained either by text or by the context frame, but not both'. It'd be interesting to see if this is indeed the case. One way to do this is to check the values of $\mu$ in Eq.4, which should be close to 0 or 1.
- Some experimental details are missing:
    - L288-290: details on how EPIC-Fields is processed.
    - L323: what does 'more difficult random trajectories' mean? how are they sampled?
    - Are the baselines re-trained in the same setting or used in a zero-shot manner?
- It'd be helpful to clarify these aspects:
    - L19-20: while textual information compensates for necessary spatiotemporal structures, it often intrudes into already observed parts of the scene
    - L25: issue of textual intrusion into observed areas
    - L109-110: override the interference of other features in video generation
    - L112-114: leverages camera movement information to further assist in clearly defining the boundaries between the text and visual elements
    - L189-191: need a control method that operates relatively independently of these features

**Questions:**

There are 3 major concerns (details above):
- Several values are missing in Tab.1 which makes it difficult to understand the trends. It'd be helpful to provide details on these missing values and EPIC-Fields experiment setting.
- The text mentions efficiency benefits in 2 places and disentangling text & image features. It'd be useful to verify if this is indeed the case.
- Several aspects of the text need further clarifications.

---

> ### Author Response · Authors · 2024-11-24
> **Official Response to Reviewer L7sN part 1**
>
> Thank you for your insightful feedback and valuable suggestions.
>
> **Clarity**: Improving the paper's readability is certainly an important aspect we aimed to address. The missing values in the table were a result of combining 4 different comparison scenarios into a single large table to save space, which inadvertently caused comprehension challenges. In the revised manuscript we have submitted, we have addressed this issue by separating the data into four individual tables. We hope this modification resolves any potential misunderstandings. The result of MotionCtrl was actually already included in the ablation study serving as a starting point. We have added these results back to Tables 3 and 4.
>
>
> **EPIC-Fields**: We didn’t evaluate T2V and I2V settings on EPIC-Fields because  the primary focus of this paper is on the image + text-to-video(T+I2V) scenario. However, there is no issue with including these comparisons. The results for T2V and I2V on EPIC-Fields are provided in the tables below. EgoSim outperforms all baselines on all metrics, demonstrating the superior performance of SEAL.
>
> | T2V| TransError | RotErr | FVD|
> |--------------------|------------|------------|------------|
> | MotionCtrl    | 15.92     | 1.78   | 991.6 |
> | CameraCtrl    | 15.72      | 1.76    | 973.5 |
> | EgoSim            | 12.88      | 1.30    | 887.2 |
>
> | I2V| TransError | RotErr | FVD   | SSIM  |
> |--------------------|------------|--------|-------|-------|
> | MotionCtrl (SVD)               | 14.82      | 1.62   | 895.1 | 0.640 |
> | CameraCtrl (SVD)    | 14.73      | 1.62   | 887.3 | 0.635 |
> | EgoSim (SVD)          | 10.75       | 1.18   | 739.0 | 0.711 |
>
> **CameraCtrl + SparseCtrl**: Yes, you are correct. We mentioned this in lines 426–429, where we stated that we directly used the publicly available code and weights for CameraCtrl + SparseCtrl. However, it failed to generate reasonable video results. As a result, the COLMAP optimization does not converge. You can view some visual examples [1](https://egosim.github.io/EgoSim/videos/failure/failure1.gif) and [2](https://egosim.github.io/EgoSim/videos/failure/failure2.gif) here to observe the failure. In these GIFs, the upper part shows videos generated by CameraCtrl + SparseCtrl, while the lower part represents the ground truth.
>
> **Efficiency**:
> 1. Sparsity: Introducing the epipolar attention mask effectively enforces sparsity, reducing unnecessary computation and storage, thereby significantly lowering GPU memory consumption. Quantitatively, with a batch size of 2 on each A100, the memory consumption without the sparsity-optimized algorithm is 78GB, whereas with the optimization, it is reduced to 42GB.
> 2. Pixel unshuffle:  This statement is based on a comparison with CameraCtrl, which employs a CNN encoder to downsample the Plücker embedding. However, we observe that the choice of a CNN encoder has no significant impact on the results. Consequently, a simpler pixel shuffle operation proves to be sufficient. Since the additional computation from CNNs is removed, it naturally saves a small portion of computation.
>
> **Values in Eq.4**:  Thank you for providing such helpful suggestions! We plotted the [distribution of sigmoid values](https://egosim.github.io/EgoSim/videos/others/sigmoid.png) from different layers, and as you pointed out, they are indeed primarily concentrated around 0 or 1. This confirms the effectiveness of the CI2V-Adapter module.
>
> **Details**:
> 1. Data preprocessing: These details are covered in Appendix A1.1. For Epic-Field, we use 611 scenes for training and 88 scenes for testing. We found that a large number of videos in Epic-Field suffer from significant motion blur due to rapid movement, which is detrimental to model learning. Therefore, we applied NAFNet to perform motion deblurring on all frames. During the training, we use a sample stride of 4 for Epic-Field.
> 2. Trajectory: These details are covered in Appendix A1.3.  We sorted the translation and rotation of the real camera poses from two datasets, and then randomly selected poses from the top 20% to 40% and 40% to 60% ranges to classify them into hard and medium groups, respectively.
> 3. Training: In the case of Epic-Field, we re-trained all baselines.

---

> ### Author Response · Authors · 2024-11-24
> **Official Response to Reviewer L7sN part 2**
>
> **Clarification**: We are happy to provide further clarifications as needed. Overall, our analyses are entirely based on the failure cases we observed in baseline models. The design improvements in EgoSim were also derived from these observations and analyses.
>
> Starting with the camera control component, mainstream methods like CameraCtrl and MotionCtrl typically integrate camera information directly into the intermediate features. In this setup, the model does learn the correlation between camera motion and video generation. However, the model at the same time learns correlations between other conditions and video generation as well. For instance, when the input is an image of a corridor as the initial frame, the training data related to corridors only involve moving forward or backward. This creates a correlation between the corridor and forward/backward motion. When these conditions are fused together through addition, the model finds it challenging to discern which correlation should take the dominant role.
>
> The failure cases we generated further validate this claim: videos generated by baseline methods struggle to produce **out-of-distribution camera movements**. When other correlations conflict with camera motion control, we naturally want camera motion to **take the highest priority and override the other correlations**. To achieve this, we need a control mechanism that is not entangled with the intermediate features but operates **independently** of them.
>
> Meanwhile, the dynamic factors, initially governed by text and image conditions, become unstable with the inclusion of camera information. This instability often manifests as abnormal dynamics, most notably a sudden and abrupt transition from the initial frame to a text-driven, imaginary scenario.
>
> To address this issue, it is essential to integrate camera information into the dynamical control modules of video generation. Regarding the integration approach, we can observe that as the camera moves, part of the scene can find reference from the initial frame, while the rest corresponds to unobserved regions. These unobserved areas can be extrapolated based on previously seen content or supplemented with additional structure provided by the text.
>
> For example, consider a scenario where the camera rotates 30° to the left and reveals a bed. The right side of the future video frame remains consistent with what was previously observed in the first frame, while the left side is generated under the guidance of the text, which specifies the bed. **This shows that different regions are either explained by the context frame (the first frame) or by the text.** The combination of camera motion and the model's understanding of 3D depth in the scene **effectively informs the model about which parts are explained by the context and which are explained by the text.** Therefore to implement this, we introduce a **sigmoid function** to control this process, enabling a smooth and dynamic fusion of context and text-driven generation.

---

> > ### Comment · Reviewer_L7sN · 2024-11-25
> > **Thanks for the clarifications**
> >
> > I appreciate the additional clarifications provided by the authors. I have raised my score.

---

> > > ### Author Response · Authors · 2024-12-01
> > >
> > > Thank you for your positive feedback! We're glad to know that clarifications were helpful and your concerns have been addressed.
> > >
> > > Once again, we deeply appreciate the time and effort you dedicated to reviewing our paper and are truly grateful for your constructive comments and suggestions!

---

### Official Review · Reviewer_SwtR · 2024-11-03

**Soundness:** 4
**Presentation:** 3
**Contribution:** 3
**Rating:** 6
**Confidence:** 3

**Summary:**

This paper tackles video generation with multi-modal condition signals: text, image, and camera pose. It introduces several model designs including employing epipolar attention to the spacetime domain for precise camera motion control and a CI2V adaptor that balances text and vision guidance based on camera information. Further, it repurposes the EPIC Fields dataset as the new dynamic scene dataset with camera annotations. Extensive experiments show the effectiveness of each proposed module.

**Strengths:**

+ The multi-modality control of video generation is an interesting topic. The proposed method fills in the gap in precise camera-conditioned image-to-video generation.
+ Most of the past camera-conditioned video generations trained on static 3D scene datasets, i.e. Realestate, DL3DV. The proposed method provides an effective practice to repurpose video understanding benchmarks for generation and to some extent shows a way to resolve the data scarcity of dynamic scene data with camera pose annotations.
+ Balancing different control signals is an intuitive challenge in multi-modal guided video generation. The proposed CI2V adapter is a simple and effective strategy to handle it.

**Weaknesses:**

- Given that Realestate is a large-scale dataset with 100 times the number of scenes compared to Epic-Field, how do you prevent overfitting your generations to static scenes?
- It would be interesting to compare the proposed method with Viewcrafter [M1] in terms of the preciseness of camera controls in a static 3D scene.
- The camera trajectories in the results are quite simple and mostly object-centric, it would be better to infer with longer, more complex trajectories in open scenes.
- [Minor] The examples in 'Interacting with the World' contain too many noticeable artifacts, e.g., hand disappearance, hand merging into objects, etc.

[M1] Yu, Wangbo, et al. "Viewcrafter: Taming video diffusion models for high-fidelity novel view synthesis." arXiv 2024.

**Questions:**

Please refer to the weaknesses section. Overall I think the task addressed is important and interesting. Most of the simple cases look fine. I suggest the authors add more comparisons with more recent methods, i.e., viewcrafter.

---

> ### Author Response · Authors · 2024-11-24
> **Official Response to Reviewer SwtR**
>
> Thank you for your insightful feedback and valuable suggestions.
>
> **Imbalanced data**:  In Appendix A1.1, we mention that although the RealEstate dataset contains more videos, the video durations in the Epic-Field dataset are significantly longer. In fact, the total amount of usable training data, measured in the number of frames, is in the range of 10 million for both datasets. Therefore, the number of training samples that can be constructed from each dataset is comparable in scale. When training the version of our model using both datasets, we sampled training clips at a 50:50 ratio from each dataset.
>
> **ViewCrafter**: ViewCrafter is indeed a highly relevant and very impressive work, and we will cite it in the related work section. However, directly comparing the preciseness of camera controls is somewhat unfair and infeasible for now for the following reasons:
> 1. ViewCrafter utilizes an additional pretrained dense stereo model DUST3R to estimate the point cloud. As such, its 3D world modeling is essentially a reflection of the dense stereo model's capabilities, rather than being extracted from video pretraining.
> 2. The camera trajectories used in the two approaches are different. Our method assumes an ego-centric exploration trajectory, whereas ViewCrafter follows an object-centric trajectory. As a result, the camera poses in these two works are defined using different coordinate systems and scales. Since their current open-source code does not cover how they handle camera poses from RealEstate, it is currently difficult to directly convert camera poses from one to another, making effective comparison infeasible. We attempted to directly use the c2w matrix from Viewcrafter; however, we found that despite both methods using the same c2w matrix as input, the resulting camera motions were unmatched. This confirms that the two approaches rely on different camera coordinate systems.
> 3. Although RealEstate may seem to feature only static scenes, as mentioned in lines 92–96, our approach enables the imagination of unobserved parts using supplementary text prompts—an ability that ViewCrafter lacks. This highlights that our task differs slightly from traditional novel view synthesis. We can illustrate this point with the following example. [Case 1](https://egosim.github.io/EgoSim/videos/viewcrafter/v0.mp4) was generated by ViewCrafter. [Case 2](https://egosim.github.io/EgoSim/videos/viewcrafter/e1.mp4) and [Case 3](https://egosim.github.io/EgoSim/videos/viewcrafter/e00.mp4) were generated by EgoSim. In [Case 2](https://egosim.github.io/EgoSim/videos/viewcrafter/e1.mp4), we specified that the model should generate plants on the left side, while in [Case 3](https://egosim.github.io/EgoSim/videos/viewcrafter/e00.mp4), we specified the generation of a floor-to-ceiling window.  As a comparison, in [Case 1](https://egosim.github.io/EgoSim/videos/viewcrafter/v0.mp4), ViewCrafter cannot create meaningful content on the left side. Note that the camera trajectories used on these two methods are not aligned.
> 4. Finally, ViewCrafter was uploaded to arXiv in September, and according to the updated [ICLR reviewer guidelines](https://iclr.cc/Conferences/2025/ReviewerGuide), comparisons are only required with works available before July 1.
>
> **Input trajectory**: The input camera trajectory is egocentric rather than object-centric, and we actually showcase some generated videos with complex and extended trajectories on the project page.
> 1. Longer Sequence: We will first explain how to generate longer videos. Specifically, this is achieved by taking the last frame of a generated video as the first frame for the next generation step. By repeating this process, longer videos can be produced. Interactive demo video [10-1](https://egosim.github.io/EgoSim/videos/interactive/s8/s8-kitchen.mp4), [3-4](https://egosim.github.io/EgoSim/videos/interactive/s5/realestate_kitchen_leftarea.mp4) are examples of this case.
> 2. Complex Trajectory: We also provide examples of complex and unordered trajectories. For instance, in interactive demo videos 5-3 and 5-4, complex trajectories are showcased. The camera trajectory in [5-3](https://egosim.github.io/EgoSim/videos/interactive/s3/realestate_bedroom_window.mp4) follows an upward, rightward spiral, while in [5-4](https://egosim.github.io/EgoSim/videos/interactive/s3/realestate_bedroom_bedstand.mp4), it moves forward in a zigzag pattern.
>
> **Artifacts**: Thank you for pointing out the limitations of the current model. This aspect of generation capability is primarily constrained by the generative power and precision of the pretrained weights we used. We believe that when our method is applied to more powerful video pretraining models, it will achieve more impressive and more consistent results.

---

> > ### Comment · Reviewer_SwtR · 2024-11-26
> >
> > I appreciate the clear responses from the authors. My concerns are addressed in the rebuttal, therefore I intend to keep the rating.

---

> > > ### Author Response · Authors · 2024-12-01
> > >
> > > Thank you for your positive feedback! We're delighted to hear that your concerns and questions have been thoroughly addressed.
> > >
> > > We sincerely appreciate the time and effort you invested in reviewing our paper and are genuinely grateful for your valuable comments and suggestions!

---

### Official Review · Reviewer_4w3N · 2024-11-07

**Soundness:** 3
**Presentation:** 1
**Contribution:** 3
**Rating:** 6
**Confidence:** 3

**Summary:**

This paper presents a novel video diffusion architecture capable of handling multiple conditioning inputs, including image, text, and camera poses, in a unified framework. The work makes meaningful technical contributions to controllable video generation, though there are areas where clarity could be improved.

**Strengths:**

The authors integrate multiple conditioning types in a coherent framework, demonstrating a novel adaptation of epipolar attention from 3D generation to video generation. The introduction of the SEAL and CI2V-adapter shows thoughtful consideration of the challenges in multi-modal video generation. The evaluation demonstrated on both static (RealEstate) and dynamic (Epic-Field) datasets, supported by both qualitative and quantitative improvements over existing methods. The extensive ablation studies further strengthen the technical contributions.

**Weaknesses:**

The paper suffers from several clarity issues that should be addressed. The experimental setup and results sections (3.2, 3.3) lack clear organization and the data preparation is not detailed throughout the paper, making it difficult to fully understand the implementation.

**Questions:**

1.1 What is the difference between EgoSim (SVD) and EgoSim in Table 1?

1.2 For Epic-Field experiments, what are the input conditions (text, image, or both) in Table 1?

1.3 Regarding LoRA usage (Line 437): Why was an additional LoRA necessary for Epic-Field when the model was already fine-tuned? This seems redundant and needs justification. Did the authors use LoRA to fine-tune the pre-trained model?

1.4 How were camera poses obtained/annotated for RealEstate and Epic-Field datasets?

1.5 How is the training and testing dataset split? What are the respective dataset sizes?

1.6 No training detail is provided. Please mention your setup such as earning rates, optimization parameters, batch sizes, number of training iterations, hardware specifications, and training time.

2. How is K* and V* calculated in practice?

3.1 L265 "attachted"" revise my review for this paper.

These clarifications would significantly improve the paper's reproducibility and technical clarity.

---

> ### Author Response · Authors · 2024-11-24
> **Official Response to Reviewer 4w3N**
>
> Thank you for the constructive comments and valuable feedback. We sincerely appreciate the reviewers for pointing out that there is room for improvement in the readability and clarity of our paper.
>
> **Clarity**: We begin by providing a detailed overview of our experimental design. The experiments are organized into three distinct scenarios: (1) camera + text-to-video (Section 3.2.1), (2) camera + image-to-video (Section 3.2.2), and (3) camera + image + text-to-video (Sections 3.2.3 and 3.3). The first two scenarios were chosen to align with the specific capabilities of the baseline models, allowing for a fair comparison. In Sections 3.2.1 and 3.2.2, we evaluate our model against these baselines, showcasing its superior performance.
> However, the primary focus of this paper is on the camera + image + text-to-video scenario. In Sections 3.2.3 and 3.3, we conducted experiments on two different datasets, RealEstate and Epic-Field, respectively. EgoSim demonstrated more precise camera control and more effective scene interaction in these cases. The description of the experimental design has been included in the revised manuscript.
>
>
> Next, we will answer each of the individual questions.
> 1. EgoSim is compatible with various pretrained models. In the case of camera+ image-to-video (I2V), we can use SVD instead of AnimateDiff.
> 2. We input camera poses, image and text for Epic-Field experiments for camera + image + text-to-video scenario.
> 3. Following the same protocol as CameraCtrl, we trained an additional LoRA, as both our work and CameraCtrl use AnimateDiff as the pretrained weights.
> 4. Both datasets inherently include camera pose information. We improved the Epic-Field dataset because it was originally unsuitable for training video diffusion models. Its videos contain extensive motion blur and lack detailed captions.
> 5. For the RealEstate dataset, we use 67,477 scenes for training and 7,289 scenes for testing. For Epic-Field, we use 611 scenes for training and 88 scenes for testing. It should be noted that the duration of each scene of Epic-Field is much longer than that in RealEstate. That’s why the number of scenes in Epic-Field is much smaller. These details are mentioned in Appendix A1.1.
> 6. In the case of SVD and AnimateDiff, we keep the original parameters of our base models fixed and only optimize the newly introduced layers and their subsequent layers using the AdamW optimizer with learning rate 2 × 10−4 . All models are trained on 8 NVIDIA A100 GPUs for 300k iterations using an effective batch size 32. We use BF16 precision for training SVD. These details are covered in Appendix A1.2.
>
> **Computation**: In practice, for each query, we can compute an epipolar attention mask for keys and values. The physical meaning of this mask is that each query essentially represents a ray in 3D space, and this ray's projections in other views are captured as epipolar attention masks. The mask values are represented as 0 and 1, where 0 indicates positions that do not need to be attended to, and 1 indicates positions that the projection passes through or is near. So  K* and V*  refer to keys and values whose mask values are 1s.
>
> **Typo**: Thank you for pointing out this typo. It has been corrected in the revised version.

---

> > ### Comment · Reviewer_4w3N · 2024-11-25
> > **Response to the authors**
> >
> > I appreciate the authors' clarification and detailed explanation. They address my main concerns and questions about this paper. I will keep the score.

---

> > > ### Author Response · Authors · 2024-12-01
> > >
> > > Thank you for your positive feedback! We're pleased to hear that the clarifications were helpful and that your concerns have been resolved.
> > >
> > > We deeply appreciate the time and effort you dedicated to reviewing our paper and are truly grateful for your insightful comments and suggestions!

---

### Meta-Review · Area_Chair_Z2aA · 2024-12-22

**Metareview:**

The paper introduces a novel video diffusion architecture for controllable video generation conditioned on multiple modalities, including text, images, and camera poses. Specifically, it introduces (i) a spacetime epipolar attention layer that can support better control of camera motion, and (ii) a CI2V adapter to balance text and vision guidance. To obtain camera annotations, the authors further repurpose video understanding dataset (Epic-Field) for the task of video generation. While there are some concerns regarding the clarity of the paper and experimental setups, such as camera trajectories used in experiments are overly simple and object-centric, limiting generalizability to open, complex scenes, or comparisons with more recent methods are missing. The authors have addressed most of them during the rebuttal phase. After careful discussion, the ACs agree the pros outweigh cons. The ACs urge the authors to incorporate the feedbacks from the reviewers into the final version of the paper. The ACs recommend acceptance.

**Additional Comments On Reviewer Discussion:**

Most of the concerns are about lack of clarity and critical experimental details, which impede reproducibility. The authors address them adequately. The authors also shared several examples of more complex and longer trajectories.

---

### Decision · Program_Chairs · 2025-01-22

Accept (Poster)